# Clinical Significance of Preoperative Inflammatory Markers in Prediction of Prognosis in Node-Negative Colon Cancer: Correlation between Neutrophil-to-Lymphocyte Ratio and Poorly Differentiated Clusters

**DOI:** 10.3390/biomedicines9010094

**Published:** 2021-01-19

**Authors:** Giulia Turri, Valeria Barresi, Alessandro Valdegamberi, Gabriele Gecchele, Cristian Conti, Serena Ammendola, Alfredo Guglielmi, Aldo Scarpa, Corrado Pedrazzani

**Affiliations:** 1Unit of General and Hepatobiliary Surgery, Department of Surgical Sciences, Dentistry, Gynecology and Pediatrics, University of Verona, 37134 Verona, Italy; giulia.turri89@gmail.com (G.T.); alessandro.valdegamberi2@aovr.veneto.it (A.V.); gabrielegecchele11@gmail.com (G.G.); cristian.conti1988@gmail.com (C.C.); alfredo.guglielmi@univr.it (A.G.); 2Department of Diagnostics and Public Health, Section of Pathology, University of Verona, 37134 Verona, Italy; valeria.barresi@univr.it (V.B.); serena.ammendola88@gmail.com (S.A.); aldo.scarpa@univr.it (A.S.)

**Keywords:** colon cancer, poorly differentiated clusters, prognostic factors, inflammatory markers, histopathological markers, immune system

## Abstract

Although stage I and II colon cancers (CC) generally show a very good prognosis, a small proportion of these patients dies from recurrent disease. The identification of high-risk patients, who may benefit from adjuvant chemotherapy, becomes therefore essential. We retrospectively evaluated 107 cases of stage I (*n* = 28, 26.2%) and II (*n* = 79, 73.8%) CC for correlations among preoperative inflammatory markers, histopathological factors and long-term prognosis. A neutrophil-to-lymphocyte ratio greater than 3 (H-NLR) and a platelet-to-lymphocyte ratio greater than 150 (H-PLR) were significantly associated with the presence of poorly differentiated clusters (PDC) (*p* = 0.007 and *p* = 0.039, respectively). In addition, H-NLR and PDC proved to be significant and independent survival prognosticators for overall survival (OS; *p* = 0.007 and *p* < 0.001, respectively), while PDC was the only significant prognostic factor for cancer-specific survival (CSS; *p* < 0.001,). Finally, the combination of H-NLR and PDC allowed an optimal stratification of OS and CSS in our cohort, suggesting a potential role in clinical practice for the identification of high-risk patients with stage I and II CC.

## 1. Introduction

Colorectal cancer (CRC) is one of the most frequent malignancies in the Western population [1]. Prognosis is mainly influenced by the completeness of surgical resection and pathological stage [2,3,4]. However, some histopathological and molecular features may play a relevant role in the definition of long-term outcomes [5] in patients affected by this neoplasia. The identification of additional prognostic factors, able to distinguish high-risk from low-risk patients, is particularly relevant in case of node-negative disease. Indeed, the use of adjuvant chemotherapy in these patients is controversial in view of the overall good prognosis [6].

In addition to tumor stage, the host immune system may play an important role in tumor development and progression [7,8,9]. Some inflammatory markers, expression of an imbalanced immune reaction, have been evaluated as prognostic factors in cancer patients [8,10,11]. In particular, neutrophil-to-lymphocyte ratio (NLR) has been tested in oncological patients [12,13,14], and its prognostic value was also suggested in patients with CRC. Various studies investigated the correlation between preoperative NLR and overall as well as disease-free survival in CRC, suggesting different threshold values and results. Most of the studies concluded that elevated NLR was associated with worse outcomes in patients with both localized and metastatic CRC [10,15,16,17,18,19,20,21,22,23]. Similarly, platelet-to-lymphocyte ratio (PLT) [23,24,25] and platelet count (PC) could predict long-term outcomes in patients with CRC [11,26,27].

Among the histopathological factors, the presence of lymphatic or vascular invasion, poor differentiation according to the World Health Organization (WHO) grading system and tumor budding are currently considered indicators of worse prognosis in stage II CRC [28,29,30]. More recently, the presence of poorly differentiated clusters (PDCs) [31] has gained attention in view of its significant correlation with higher recurrence risk and shorter overall survival in patients with stage II CRC [32,33,34]. Nevertheless, the assessment of PDC is rarely adopted for prognostic stratification in routine clinical practice, as conventional tumor grading system is still preferred in the AJCC guidelines [35].

This study aimed at evaluating the potential correlations between preoperative inflammatory biomarkers and histopathological characteristics in node negative colon cancer, as well as their impact on long-term prognosis.

## 2. Materials and Methods

### 2.1. Inclusion Criteria and Population under Study

The original population under study consisted of all patients undergoing surgery for CRC (*n* = 1418) at the Division of General and Hepatobiliary Surgery, University of Verona Hospital, between January 2005 and December 2015 The inclusion criteria were: age ≥ 18 years; histology-proven colon cancer; absence of nodal or distant metastasis (AJCC/UICC TNM Stage I and II); availability of histological slides or paraffin block of the primary tumor; data on preoperative NLR, PLR and PC; and absence of residual disease after surgery (R0 resection). Patients with rectal cancer were excluded from the analysis.

### 2.2. Assessment of Inflammatory Markers

Neutrophil count, lymphocyte count and PC were obtained from venous blood within 2 weeks before the date of surgery. NLR and PLR were calculated by dividing the absolute number of neutrophils or platelets by the absolute number of lymphocytes, respectively. Blood samples were drawn by an expert phlebotomist in vacuum blood tubes containing K2-EDTA (Terumo Europe NV, Leuven, Belgium). The complete blood cell count (CBC) was performed using Advia 2120 (Siemens Healthcare Diagnostics, Tarrytown NY, USA). The local reference ranges are 150–400 × 10^9^/L for platelets, 4.3–10.0 × 10^9^/L for total white blood cells (WBC), 2.0–7.0 × 10^9^/L for neutrophils and 0.95–4.5 × 10^9^/L for lymphocytes. The same analyzer was used throughout the study period. The quality and comparability of test results were validated by data of both internal quality control (IQC) and external quality assessment (EQA) [36].

### 2.3. Histological Evaluation

All cases included underwent histopathological revision as previously described in detail [32]. Briefly, hematoxylin–eosin-stained histological slides were revised to assess the depth of infiltration (pT1, pT2, pT3 and pT4) and histological grading according to the WHO criteria, lympho-vascular invasion (LVI), perineural invasion (PNI), tumor budding, presence of inflammation and PDC count. PDC were defined as clusters of at least 5 tumor cells lacking a glandular structure, at the invasive front or in the tumor stroma and counted in one hot spot under the microscopic field of ×20 objective lens (i.e., a microscopic field with a major axis of 1 mm). The 8th Edition of the American Joint Committee on Cancer (AJCC) and the Union International Contre Le Cancer (UICC) criteria were used for reporting the pathology specimens [35].

### 2.4. Preoperative Work-Up and Surgical Technique

All patients were staged with preoperative colonoscopy, chest-abdomen-pelvis computed tomography (CT) and carcinoembryonic antigen (CEA) measurement. In the case of dubious hepatic lesions, magnetic resonance was used to clarify the preoperative staging.

The main goal of surgery was the complete excision of the cancer burden in order to obtain an R0 resection. The extent of the resection was planned according to cancer location, disease stage and patient’s general conditions. Anatomical resections with ligation of vessels at their origin were the procedures of choice in order to achieve an adequate lymphadenectomy [37].

### 2.5. Data Collection and Statistical Analysis

Data were extracted from a prospectively maintained database. Demographic, clinical, surgical, hematological and histopathological variables were analyzed. All methods used in this study were performed in accordance with the relevant ethical guidelines and regulations of the University Hospital of Verona, where the investigation was carried out. The study was approved by the Verona University Hospital Ethics Committee (09/07/2016, ID number: 42763-CRINF-1034 CESC). Informed consent was obtained from all patients enrolled in the study. On preliminary analysis, preoperative NLR, PLR and PC were found to be normally distributed. The optimal cut-off values for NLR (≥3) [19,20], PLR (≥150) [25] and PC (≥350 × 10^9^/L) [10] as dichotomous predictors of survival were chosen based on previously published literature. The correlation between preoperative inflammatory markers and pathological features was investigated using independent t test or Mann–Whitney U test for continuous variables and chi-squared test or Fisher’s exact test for categorical variables, as appropriate. Continuous data were reported as mean (+SD) or median (range) as appropriate according to distribution, while categorical data were reported as numbers and percentages.

Survival and follow-up data were obtained by collecting outpatient clinical records or by directly contacting the patient or their relatives. The median length (range) of follow up was 104 (3–160) months considering the whole population and 113 (76–160) months considering surviving patients only. At the time of analysis, 75 patients had completed their follow-up and 32 have died.

Survival analysis was computed using the Kaplan–Meier method and compared by the log-rank test, with time of overall survival (OS) measured from the date of surgery to the date of death from any cause or most recent follow-up and cancer-specific survival (CSS) as months from the date of surgery to the date of death from cancer. Multivariate analysis was performed by Cox regression model taking into account clinical and pathological characteristics and inflammatory markers that were found to significantly influence long-term survival on univariate analysis.

All statistical tests were two-sided, and association were considered statistically significant at a nominal level of 0.05 (*p* < 0.05). Statistical analysis was performed using SPSS (version 23, SPSS, Chicago, IL, USA).

## 3. Results

In total, 107 patients fulfilled the inclusion criteria and were included in the analysis (Figure 1). Twenty-eight (28) tumors (26.2%) were classified as TNM stage I and 79 (73.8%) as stage II. None of the patients received adjuvant chemotherapy.

Table 1 reports the correlation between inflammatory markers and clinical-pathological variables. Forty-five patients (42.1%) had an NLR value greater than 3 (H-NLR). H-NLR was significantly associated with serosal invasion (31% vs 11.3%; *p* = 0.036) and presence of PDC (51.1% vs 24.2%; *p* = 0.007). A PLR value greater than 150 (H-PLR) was associated with a significantly higher rate of mucinous histotype (18.8% vs 4.6%; *p* = 0.042) and presence of PDC (43.7% vs 23.2%; *p* = 0.039). Finally, a PC greater than 350 × 10^9^/L (H-PC) was associated with a higher rate of right-sided CC (76.7% vs 39%; *p* = 0.001), mucinous histotype (30% vs 6.5%; *p* = 0.003) and poorly differentiated (G3) tumors (23.3% vs 2.6%; *p* = 0.002). No significant associations were demonstrated between NLR, PLR and PC values and the amount of inflammatory reaction, nor with lympho-vascular, perineural invasion or tumor budding.

At survival analysis, NLR, among inflammatory markers and PDC, among histopathological factors, demonstrated to significantly and independently influence OS and CSS (Figure 2 and Table 2).

Accordingly, the combined effect on long-term outcomes of NLR and PDC was evaluated. As shown in Figure 3, excellent long-term outcomes were observed in PDC negative cases almost independently from NLR values. Conversely, long-term survival demonstrated to be negatively influenced by the presence of PDC, with a significantly worse prognosis in H-NLR cases, both considering OS (*p* < 0.001) and CSS (*p* < 0.001).

In the Cox regression multivariate analysis, age above the median (*p* < 0.001), TNM stage II (*p* = 0.035), H-NLR (*p* = 0.007) and the presence of PDC (*p* < 0.001) were independent predictors of shorter OS. Presence of PDC was the only independent prognostic factor for shorter CSS (*p* < 0.001), although H-NLR was nearly significantly associated (*p* = 0.072) (Table 3).

The Cox regression multivariate analysis for OS and CSS was also conducted using the combination of PDC and NLR. The presence of both negative prognostic factors showed an additive effect; the HR for PDC-present/L-NLR was 19.91 (2.14–185.11) compared to an HR of 56.67 (95% CI 6.63–483.94) for PDC-present/H-NLR (*p* < 0.001) (Table 4).

## 4. Discussion

In the current study, we analyzed the association and the prognostic role of preoperative inflammatory markers and the main histopathological features in surgically resected CCs in the absence of lymph node metastases. The main results of this study are: (1) H-NLR values are significantly associated with the presence of PDCs; (2) both H-NLR and PDC confirmed to be significant and independent survival prognosticators; and (3) the combination of NLR and PDC allows a better stratification of OS and CSS in TNM Stage I and II colon cancer. To the best of our knowledge, this is the first study demonstrating a correlation between preoperative inflammatory markers and the presence of PDCs in patients with CC.

According to current guidelines [28,29,30], adjuvant chemotherapy in stage II CC is considered only for patients with specific risk factors, namely serosal infiltration (pT4), presence of lymphatic or vascular invasion and fewer than 12 analyzed nodes. However, among node-negative CCs, 5% of stage I and 12% of stage II tumors will develop a recurrence within five years from surgery [6]. Some molecular parameters, such as microsatellite instability and KRAS/BRAF mutations, have been associated with survival outcomes [5,38,39], however their assessment requires sophisticated and expensive techniques. Therefore, great interest has been directed towards the identification of some readily available and inexpensive markers which could be useful for the detection of patients who may benefit from systemic chemotherapy. Since neutrophils, platelets and lymphocytes are routinely measured as part of the preoperative work-up of patients undergoing surgery, their possible prognostic value could be very relevant in clinical practice.

The clinical impact of inflammatory markers has been partially confirmed in our study. Although NLR showed a significant association with increased risk of death from cancer (Table 2), and it was an independent prognostic variable for shorter OS at the multivariate analysis (Table 3), PLT and PC did not demonstrate any relevant association with long-term outcomes. This is in accordance with previously published studies [10,22,23,40]. Although other studies concluded that NLR is an important inflammatory biomarker in CRC, several issues should be remarked. In the study by Li and colleagues [21] on 5336 patients with CRC, which is largest published series, H-NLR was an independent prognostic factor for OS at multivariate analysis. However, the significance of NLR and other inflammatory markers in patients who did not undergo adjuvant chemotherapy was not demonstrated; this is in line with our results. Likewise, Haram at al. [22] conducted a systematic review to assess the prognostic role of NLR in metastatic and non-metastatic CRC. They concluded that preoperative NLR > 5 was associated with poorer overall survival in patients with CRC, but no association was found with the other chosen cut-offs. Malietzis et al. [20] did not identify an independent prognostic role of H-NLR (>3) in 506 patients with non-metastatic CRC who did not receive adjuvant chemotherapy. Finally, the systematic review and meta-analysis by Zhang et al. [23] found a significant association between NLR > 5, PLR > 150, PC > 400 and overall survival. However, none of the study evaluated cancer-specific survival. Furthermore, most of the studies included colon as well as rectal cancer [41,42], therefore producing results that may be biased because of the difference in treatment and prognosis of the two locations.

With regards to histopathological markers of poor prognosis, the presence of PDCs has recently gained attention as a promising prognostic factor in patients with CRC [33,34]. They reflect tumor de-differentiation, and their evaluation on hematoxylin and eosin-stained slides is more reproducible than WHO grading [32,43]. No previous study evaluated the association between PDC and inflammation-based scores in CC. This study is the first to show a significant association between H-NLR and PDC, and their cumulative negative effect on OS and CSS. The presence of an imbalanced inflammatory response measured on peripheral blood may reflect the presence of a more de-differentiated and aggressive disease. A previous study reported a significant correlation of tumor budding with preoperative neutrophil count, but not with NLR [44]. Although tumor budding and PDC have morphological and immunohistochemical similarities and might both represent tumor de-differentiation [45,46], there is not a clear evidence that they biologically overlap.

In patients with TNM stage I and II CC, inflammatory markers may permit preoperative identification of high-risk patients, whereas pathological markers lead to postoperative stratification of patients with a reduced survival probability and a higher risk of recurrence. Namely, patients with cT4 and one of these risk factors may be considered for neoadjuvant treatment, or patients with H-NLR and PDCs may receive adjuvant chemotherapy even in absence of node metastases and other risk factors. In fact, it should be noted that the association between NLR and PDC resulted to be a better prognosticator of CSS than TNM stage itself, suggesting than even TNM stage I patients with PDC (14 cases, 50%) or H-NLR (11 cases, 39.2%) may benefit from adjuvant chemotherapy. Similarly, PDC and NLR may assist in selecting endoscopically resected “early cancers” that should merit to undergo surgical resection.

The main limitation of this study relates to its retrospective nature and the limited sample size. Although we considered an initial large population, many patients were excluded due to the unavailability of all histological slides and/or hematological parameters. However, our study also has many strengths. First, the population includes homogeneous cases of node-negative CC who did not receive adjuvant chemotherapy. In fact, at their time of surgery, no adjuvant chemotherapy was indicated for TNM stage II cancers, even in the presence of pathological risk factors. This allows us to abolish the potential bias related to the administration of adjuvant chemotherapy. Second, the consistent follow-up time assured the identification of some late recurrences that may characterize the postoperative course of early stages CC.

## 5. Conclusions

Our study suggests that both NLR and PDC significantly affect survival even when limiting the analysis to stage I and II CCs. Noteworthy, we observed an increased rate of PDC positivity in patients with high values of NLR. In addition, NLR significantly and independently stratified OS and CSS in cases with PDC positivity. Further studies with a higher number of cases are required in order to confirm our observations and identify the effective clinical value of the association of H-NLR and PDC.

## Figures and Tables

**Figure 1 biomedicines-09-00094-f001:**
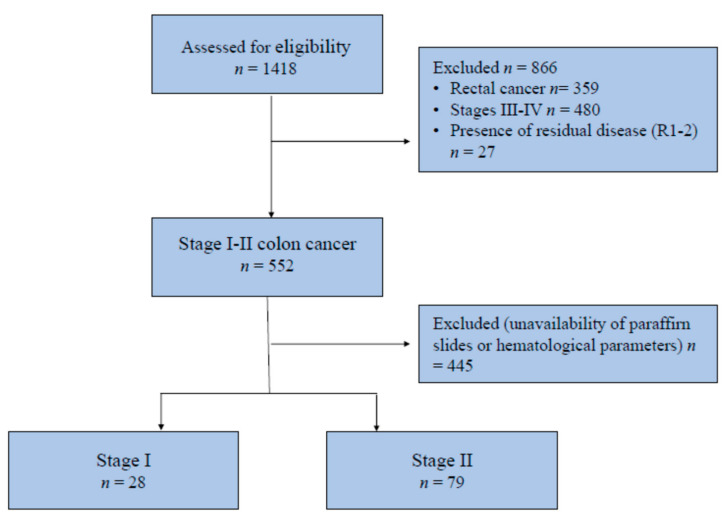
CONSORT diagram for patient inclusion.

**Figure 2 biomedicines-09-00094-f002:**
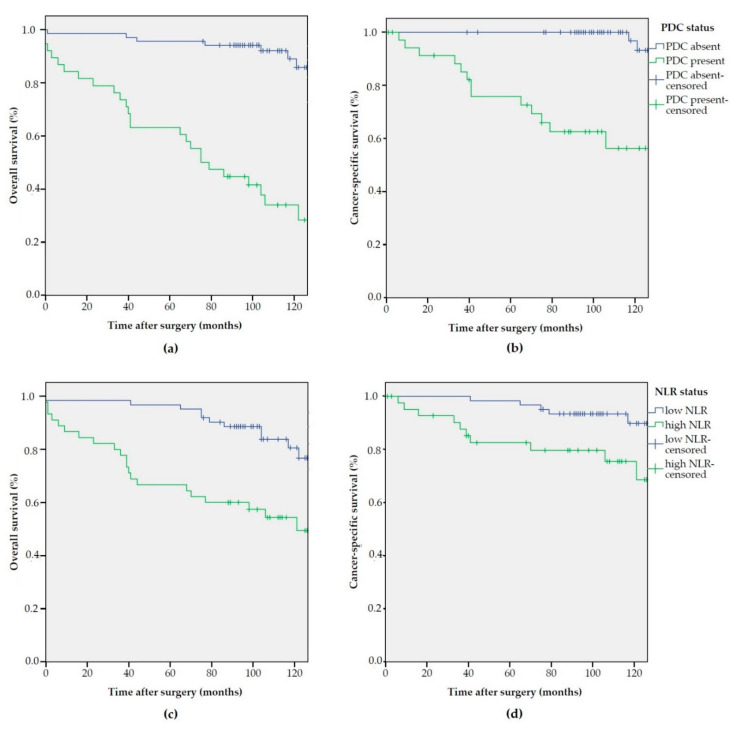
Kaplan–Meier estimates of overall survival (OS) and cancer-specific survival (CSS) according to PDC and NLR status: (**a**) OS according to PDC status (*p* < 0.001); (**b**) CSS according to PDC status (*p* < 0.001); (**c**) OS according to NLR status (*p* < 0.001); and (**d**) CSS according to NLR status (*p* = 0.011).

**Figure 3 biomedicines-09-00094-f003:**
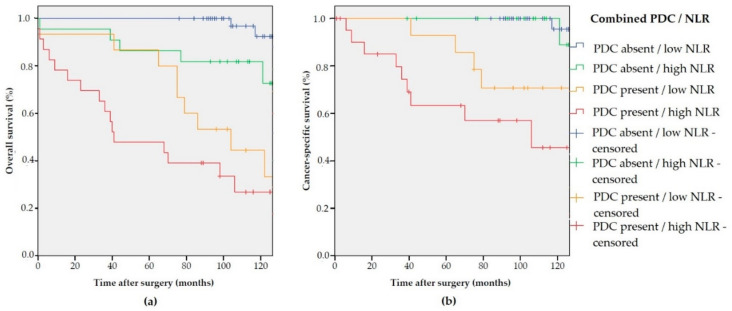
(**a**) Kaplan–Meier estimates of OS (*p* < 0.001) according to the combined PDC/NLR variable; and (**b**) Kaplan–Meier estimates of CSS (*p* < 0.001) according to the combined PDC/NLR variable.

**Table 1 biomedicines-09-00094-t001:** Correlations between NLR, PLR and PC and main clinical and pathological variables for the 107 patients under study.

Data		H-NLR (*n* = 45)	*p* Value	H-PLR (*n* = 64)	*p* Value	H-PC (*n* = 30)	*p* Value
Age, mean (SD)		70.5 (14.9)	0.238	70.8 (14.1)	0.294	68.5 (17.5)	0.846
Gender, male (%)		26 (57.7)	0.844	33 (51.6)	0.321	12 (40)	0.051
Tumor location (%)			0.331		0.238		0.001
	Right colon	25 (55.6)		35 (54.7)		23 (76.7)	
	Left colon	20 (44.4)		29 (45.3)		7 (23.3)	
Elective surgery (%)		42 (93.3)	0.307	61 (95.3)	0.647	29 (96.7)	1
CACI, mean (SD)		2.9 (1.5)	0.845	3.1 (1.6)	0.264	3.2 (2.0)	0.389
Mucinous carcinoma, *n* (%)		5 (11.1)	0.774	12 (18.8)	0.042	9 (30)	0.003
Depth of tumor invasion, *n* (%)			0.036		0.055		0.081
	pT1–2	11 (24.4)		12 (18.7)		6 (20)	
	pT3	20 (44.4)		36 (56.2)		14 (46.7)	
	pT4	14 (31.2)		16 (25.1)		10 (33.3)	
AJCC TNM Stage II, *n* (%)		34 (75.6)	0.825	51 (79.7)	0.118	23 (26.7)	0.808
Harvested lymph-nodes ≥ 12, *n* (%)		42 (93.3)	0.731	58 (90.6)	0.738	26 (86.7)	0.264
Tumor grading, high grade, *n* (%)		6 (13.3)	0.162	6 (9.4)	0.738	7 (23.3)	0.002
Inflammatory reaction, present, *n* (%)		36 (80)	0.602	52 (81.2)	0.604	25 (83.3)	1
Budding, high grade, *n* (%)		4 (8.9)	0.446	3 (4.7)	0.636	1 (3.3)	0.063
LVI present, *n* (%)		15 (33.3)	1	19 (29.7)	0.529	12 (40)	0.362
PNI present, *n* (%)		9 (20)	0.815	14 (21.9)	1	8 (26.7)	0.439
PDC present, *n* (%)		23 (51.1)	0.007	28 (43.7)	0.039	15 (50)	0.071

SD, standard deviation; CACI, Charlson Adjusted Comorbidity Index; LVI, Lymphovascular Invasion; PNI, Perineural Invasion; PDC, Poorly Differentiated Clusters. Number in parentheses are percentages, unless specified otherwise.

**Table 2 biomedicines-09-00094-t002:** Kaplan–Meier estimates of survival probability at 5 years according to main clinical–pathological variables for the 107 patients under study.

Data		Pts	OS	*p*	CSS	*p*
Age						
	≤median	45 (42.1%)	95.6%	0.009	97.7%	0.521
	>median	62 (57.9%)	75.8%		87.7%	
Gender						
	Male	60 (56.1%)	91.7%	0.562	93%	0.862
	Female	47 (43.9%)	78.7%		90.9%	
Tumor location						
	Right colon	53 (49.5%)	88.7%	0.866	94%	0.533
	Left colon	54 (50.5%)	79.6%		90.1%	
TNM Stage						
	I	28 (26.2%)	82.1%	0.015	100%	0.868
	II	79 (73.8%)	84.8%		89.5%	
Harvested lymph-nodes						
	<12	9 (8.4%)	66.7%	0.269	87.5%	0.359
	≥12	98 (91.6%)	85.7%		92.5%	
Lympho-vascular invasion						
	LVI -	72 (67.3%)	88.9%	0.135	95.6%	0.109
	LVI +	35 (32.7%)	74.3%		83.9%	
Perineural invasion						
	PNI -	84 (78.5%)	84.5%	0.548	93.6%	0.576
	PNI +	23 (21.5%)	82.6%		86.4%	
PDC						
	Absent	69 (64.5%)	95.7%	<0.001	100%	<0.001
	Present	38 (35.5%)	63.2%		75.7%	
NLR						
	L-NLR	62 (57.9%)	96.8%	<0.001	98.4%	0.011
	H-NLR	45 (42.1%)	66.7%		82.5%	
PLR				0.563		0.825
	L-PLR	43 (40.2%)	90.7%		95.2%	
	H-PLR	64 (59.8%)	79.7%		89.8%	
PC				0.457		0.894
	L-PC	77 (72%)	88.3%		93.3%	
	H-PC	30 (38%)	73.3%		88.6%	
Combined						
	PDC absent/L-NLR	47 (43.9%)	100%	<0.001	100%	<0.001
	PDC absent/H-NLR	22 (20.6%)	86.4%		100%	
	PDC present/L-NLR	15 (14%)	86.7%		92.9%	
	PDC present/H-NLR	23 (21.5%)	47.8%		63.3%	

**Table 3 biomedicines-09-00094-t003:** Multivariable analysis for overall- and cancer-specific survival.

Data		OS: HR (95% CI)	*p* Value	CSS: HR (95% CI)	*p* Value
Age			<0.001		0.169
	≤median	-		-	
	>median	5.0 (2.07–12.29)		2.27 (0.71–7.33)	
Gender			0.303		0.920
	Male	-		-	
	Female	0.67 (0.31–1.44)		0.94 (0.61–2.06)	
Tumor location			0.107		0.263
	Right colon	-		-	
	Left colon	2.05 (0.94–4.51)		1.95 (0.61–6.30)	
Stage			0.035		0.688
	I	-		-	
	II	0.43 (0.21–0.94)		1.30 (0.36–4.63)	
NLR			0.007		0.072
	L-NLR	-		-	
	H-NLR	4.25 (1.77–10.26)		4.38 (1.25–15.34)	
PDC			<0.001		<0.001
	Absent	-		-	
	Present	11.96 (4.70–30.40)		26.37 (5.30–131.28)	

**Table 4 biomedicines-09-00094-t004:** Multivariable analysis for overall- and cancer-specific survival conducted with the combination variable.

Data		OS: HR (95% CI)	*p* Value	CSS: HR (95% CI)	*p* Value
Age			0.001		0.196
	≤ median	-		-	
	> median	5.0 (2.07–12.29)		2.27 (0.71–7.33)	
Gender			0.170		0.572
	Male	-		-	
	Female	0.67 (0.31–1.44)		0.74 (0.61–2.06)	
Tumor location			0.107		0.263
	Right colon	-		-	
	Left colon	2.05 (0.94–4.51)		1.95 (0.61–6.30)	
Stage			0.04		0.688
	I	-		-	
	II	0.43 (0.21–0.94)		1.30 (0.36–4.63)	
Combined PDC/NLR			<0.001		<0.001
	PDC absent/L-NLR	-		-	
	PDC absent/H-NLR	5.78 (1.11–30.27)		2.52 (0.16–40.74)	
	PDC present/L-NLR	19.13 (3.96–92.36)		19.91 (2.14–185.11)	
	PDC present/H-NLR	43.58 (9.29–204.34)		56.67 (6.63–483.94)	

## Data Availability

The data presented in this study are available on request from the corresponding author. The data are not publicly available due to privacy reasons.

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
