# Peer review of "Clinical Significance of Preoperative Inflammatory Markers in Prediction of Prognosis in Node-Negative Colon Cancer: Correlation between Neutrophil-to-Lymphocyte Ratio and Poorly Differentiated Clusters"

_biomedicines, 2021, doi:10.3390/biomedicines9010094_

Round 1

Reviewer 1 Report

The manuscript titled: „Clinical significance of pre-operative inflammatory markers in prediction of prognosis in node-negative colon cancer. Correlation between neutrophil-to-lymphocyte ratio and poorly differentiated clusters” by Giulia Turri et al presents retrospective analysis  of the stage I and II colon cancer (CC) cases (n=107). The node-negative patients routinely do not receive post-operative  chemotherapy since their prognosis in very good, however a small group of them dies from recurrent CC. The Authors  searched for possible prognostic markers which would indicate those node-negative patients which might benefit from  chemotherapy.  Thus, such search was justified from the medical point of view.

The Authors analyzed the archived data concerning demographic, clinical, surgical, hematological and  histopathological features of the patients. All cases underwent histopathological revision, performed using  the archived histological slides. The revision included assessment of the PDC (poorly differentiated clusters) count, since the presence of PDCs has recently been correlated with higher recurrence risk and shorter survival of stage II CC patients. Also, the inflammatory markers, i. e.  neutrophil-to lymphocyte (NLR) and platelet-to-lymphocyte (PLR) ratios were calculated. Among the variables tested, they found that the  high NLR  and high PLR were significantly correlated with the presence of PDC. What seems important, high NLR and PDC were significant and independent prognostic markers for overall survival while PDC was the only significant marker for cancer-specific survival. The findings are interesting though based on rather small sample of cases and thus further  studies with higher number of cases are necessary. Nevertheless, this study shows a promising way for future investigations.

Comment:

Please, improve quality of Figures 2 and 3 – the font size of all descriptions should be increased. In present state, the letters are illegible without magnification.

Author Response

REVIEWER 1

The manuscript titled: „Clinical significance of pre-operative inflammatory markers in prediction of prognosis in node-negative colon cancer. Correlation between neutrophil-to-lymphocyte ratio and poorly differentiated clusters” by Giulia Turri et al presents retrospective analysis of the stage I and II colon cancer (CC) cases (n=107). The node-negative patients routinely do not receive post-operative chemotherapy since their prognosis in very good, however a small group of them dies from recurrent CC. The Authors searched for possible prognostic markers which would indicate those node-negative patients which might benefit from chemotherapy.  Thus, such search was justified from the medical point of view.

The Authors analyzed the archived data concerning demographic, clinical, surgical, hematological and histopathological features of the patients. All cases underwent histopathological revision, performed using the archived histological slides. The revision included assessment of the PDC (poorly differentiated clusters) count, since the presence of PDCs has recently been correlated with higher recurrence risk and shorter survival of stage II CC patients. Also, the inflammatory markers, i. e.  neutrophil-to lymphocyte (NLR) and platelet-to-lymphocyte (PLR) ratios were calculated. Among the variables tested, they found that the high NLR and high PLR were significantly correlated with the presence of PDC. What seems important, high NLR and PDC were significant and independent prognostic markers for overall survival while PDC was the only significant marker for cancer-specific survival. The findings are interesting though based on rather small sample of cases and thus further studies with higher number of cases are necessary. Nevertheless, this study shows a promising way for future investigations.

Comment:

Please, improve quality of Figures 2 and 3 – the font size of all descriptions should be increased. In present state, the letters are illegible without magnification.

Response:

Thank you very much for the kind review. As stated in the paragraph analyzing limitations of the study (page 11, line 254-255), we also feel that our results are limited by the small sample, but still relevant and interesting for further investigations including a larger number of cases.

We have increased the font size in both figures as suggested.

Reviewer 2 Report

The background must be improved with a summary of pertinent findings from additional references, including several recent systematic reviews and meta-analyses (eg. Haram et al 2017, Zhang et al 2017, Tsai et al 2016, and Mei et al 2014). The gap in knowledge is not clearly defined.

A significant limitation of the study is the relatively small samples size (n=107) and high exclusion rate (n=445). Is the included patient population representative?

The text on K-M curves (Fig 2 and 3) is difficult to read and should modified.

The discussion on the systemic inflammatory markers (ie NLR, PDR and PC) needs expansion in the context of findings from systematic reviews, meta-analyses and recent publications.

The statements “No previous study evaluated the association between PDC and inflammation-based scores in CC” and “To the best of our knowledge, this is the first study demonstrating a correlation between pre-operative inflammatory markers and the presence of PDCs in patients” are not entirely correct and should be modified. For example, Jakubowska et al 2020 reported a correlation between NLR and tumour budding. A more comprehensive search of literature is required, and the findings should be discussed in light of previously reported findings.

What is meant by the statement “The presence of an imbalanced inflammatory response measured on peripheral blood may reflect the presence of a more de-differentiated and aggressive disease”. Tumour grade (differentiation) was not reported for the cohort, so how was this conclusion made?

The statement “Finally, PDC and NLR may assist in selecting endoscopically resected “early cancers” that should merit to undergo surgical resection” needs clarification.

Reference 7 (Mazaki et al) is missing the journal name.

Author Response

REVIEWER 2

We wish to thank the reviewer for his/her comments, that were useful to improve our paper. All changes in the text have been highlighted using word revision track mode.

1. The background must be improved with a summary of pertinent findings from additional references, including several recent systematic reviews and meta-analyses (eg. Haram et al 2017, Zhang et al 2017, Tsai et al 2016, and Mei et al 2014). The gap in knowledge is not clearly defined.

Response: Haram et al. conducted a systematic review in 2017 to assess the value of NLR in predicting clinical and survival outcomes. They found eleven studies fulfilling their inclusion criteria, with 8688 patients analyzed. However, more than half of the patients came from the series by Li et al. including 5336 patients, which was in fact cited in the Discussion paragraph (page 10, lines 215-219) and that has been added to the Introduction as well (page 2, line 48). Following the reviewer suggestion, the paper by Haram was included in the reference list (page 13, lines 350-351), and its results commented in the introduction (page 2, line 48) and discussion section (page 10, lines 219-222).

The systematic review and meta-analysis from Zhang et al. evaluated the prognostic role of NLR, PLR and platelet count (PLT) in patients with stage I-IV CRC. They identified 16 studies for NLR (8691 patients), 6 studies for PLR (1113 patients), and 9 studies for PLT (3685). However, it did not include the large series from Li et al. since the literature search was conducted up to April 2016. The meta-analysis found a significant association between NLR>5, PLR>150, PLT>400 and overall survival, but no association with NLR>3. As suggested, the reference has been added to the reference list (page 13, lines 352-353), and its results commented in the introduction (page 2, line 48) and the discussion section (page 13, lines 225-227).

The systematic review and meta-analysis from Tsai et al. included 7741, and was published in 2016, before the study from Haram et al and Zhang et al. Consequently, the data from the two studies are overlapping, and it will be redundant to cite Tsai et al.

Finally, the systematic review and meta-analysis from Mei et al. does not seem relevant to the background of our study as it assessed different parameters. In their paper, Mei et al. included studies that focused on “generalised tumour inflammatory infiltrate and associated T lymphocyte subsets (including CD3þ, CD8þ, FoxP3þ, CCR7þ, CD45ROþ and GRBþ lymphocytes) in CRC patients identified by HE staining and/or IHC staining”, while our paper evaluated the neutrophil-to-lymphocyte ratio defined as the ratio between absolute number of neutrophils and lymphocyte on peripheral blood count (Materials and Methods, section Assessment of Inflammatory Markers, page 2, lines 72-73).

2. A significant limitation of the study is the relatively small samples size (n=107) and high exclusion rate (n=445). Is the included patient population representative?

Response: The retrospective nature of our study did not permit to retrieve or review the histological slides of all colorectal cancer patients treated at our center. Similarly, we could not obtain the pre-operative value of hematological parameters of all patients, as lymphocyte count was not routinely done in all patients. We acknowledge the small sample size as one of the limitations of our study, as specified in the Discussion paragraph (page 11, lines 254-255). Despite the exclusion of a large number of cases, we believe that our final population is representative of the whole cohort.

3. The text on K-M curves (Fig 2 and 3) is difficult to read and should modified.

Response: We have increased the font size in both figures as suggested.

4. The discussion on the systemic inflammatory markers (ie NLR, PDR and PC) needs expansion in the context of findings from systematic reviews, meta-analyses and recent publications.

Response: As suggested the discussion about systemic inflammatory markers (NLR, PLR, and PC) has been expanded including results from the systematic reviews and meta-analysis by Haram and Zhang (page 10, lines 219-222 and 225-227).

5. The statements “No previous study evaluated the association between PDC and inflammation-based scores in CC” and “To the best of our knowledge, this is the first study demonstrating a correlation between pre-operative inflammatory markers and the presence of PDCs in patients” are not entirely correct and should be modified. For example, Jakubowska et al 2020 reported a correlation between NLR and tumour budding. A more comprehensive search of literature is required, and the findings should be discussed in light of previously reported findings.

Response: Thank you for your comment. Actually, this is the first study assessing the correlation between PDC and pre-operative inflammatory markers in colon cancer. Although tumor budding and PDC have some similarity, currently there is no clear evidence that they biologically overlap. Following the reviewer suggestion, the paper by Jakubowska et al. was included in the reference and its results were commented in the discussion section (page 10, lines 238-242).

6. What is meant by the statement “The presence of an imbalanced inflammatory response measured on peripheral blood may reflect the presence of a more de-differentiated and aggressive disease”. Tumour grade (differentiation) was not reported for the cohort, so how was this conclusion made?

Response: Thank you for this comment. WHO histological grade was reported in this cohort, and high grade (G3) was significantly associated with H-PC. In view of the significant association between an imbalanced inflammatory response and the presence of PDC, which also reflects tumor de-differentiation, we stated that “The presence of an imbalanced inflammatory response measured on peripheral blood may reflect the presence of a more de-differentiated and aggressive disease”.

7. The statement “Finally, PDC and NLR may assist in selecting endoscopically resected “early cancers” that should merit to undergo surgical resection” needs clarification.

Response: Thank you very much for your comment. Endoscopically resected pT1 adenocarcinomas may only be treated by local excision in the absence of risk factors. On the other hand, pT1 with pathological risk factors (distance from resection margin <1 mm, presence of vascular and lymphatic invasion, high tumor grade, and high-grade tumor budding) present a 5-25% risk of lymphatic metastases and should undergo surgical resection. Similarly, we suggest that the presence of PDC or H-NLR may assist the selection of high-risk pT1 necessitating surgical resection.

8. Reference 7 (Mazaki et al) is missing the journal name.

Response: The citation has been corrected and the journal name has been included.